# Decrease in Mucosal IL17A, IFNγ and IL10 Expressions in Active Crohn’s Disease Patients Treated with High-Dose Vitamin D Alone or Combined with Infliximab

**DOI:** 10.3390/nu12123699

**Published:** 2020-11-30

**Authors:** Mia Bendix, Anders Dige, Søren Peter Jørgensen, Jens Frederik Dahlerup, Bo Martin Bibby, Bent Deleuran, Jørgen Agnholt

**Affiliations:** 1Department of Hepatology and Gastroenterology, Aarhus University Hospital, 8200 Aarhus, Denmark; andedige@rm.dk (A.D.); soerjoer@rm.dk (S.P.J.); jensdahl@rm.dk (J.F.D.); joeragnh@rm.dk (J.A.); 2Department of Public Health—Department of Biostatistics, Aarhus University, 8000 Aarhus, Denmark; bibby@ph.au.dk; 3Department of Rheumatology, Aarhus University Hospital, 8200 Aarhus, Denmark; bd@biomed.au.dk; 4Department of Biomedicine, Aarhus University, 8000 Aarhus, Denmark

**Keywords:** vitamin D treatment, Crohn’s disease, infliximab

## Abstract

Background: Vitamin D treatment may reduce Crohn’s disease (CD) activity by modulating the mucosal immune function. We investigated if high-dose vitamin D +/− infliximab modulated the mucosal cytokine expression in active CD. Methods: Forty CD patients were randomized into: infliximab + vitamin D; infliximab + placebo-vitamin D; placebo-infliximab + vitamin D or placebo-infliximab + placebo-vitamin D. Infliximab (5 mg/kg) and placebo-infliximab were administered at weeks 0, 2 and 6. Oral vitamin D was administered as bolus 200,000 international units (IU) per week 0 followed by 20,000 IU/day for 7 weeks or placebo. Endoscopy with biopsies was performed at weeks 0 and 7 where endoscopic activity was measured and mucosal mRNA cytokine expression was examined. C-reactive protein (CRP), fecal calprotectin and Harvey-Bradshaw Index (HBI) were measured at weeks 0, 2 and 6. Results: High-dose vitamin D treatment alone and combined with infliximab decreased the IL17A, IFNγ and IL10 expression. High-dose vitamin D alone did not significantly decrease the disease activity, CRP or calprotectin. Combined infliximab and vitamin D treatment was not clinically significantly superior to monotherapy with infliximab. Conclusions: High-dose vitamin D as monotherapy and combined with infliximab decreases IL17A, IFNγ and IL-10 expression in mucosa within treatment groups. This did not induce a statistically significant decreased disease activity. EudraCT no.2013-000971-34.

## 1. Introduction

Crohn’s disease (CD) is a chronically intestinal inflammation characterized by episodes with disease activity followed by periods of remission. It is challenging to maintain CD patients in remission since response to treatment is reduced over time, including standard-of-care therapy with the anti-TNFα treatment, infliximab [1]. CD patients with chronic inflammation have increased risk for development of stenosis, need for hospitalization and surgery [2]. Therefore, effective maintained treatment efficacy is of great importance. In CD, inflammation is dominated by an unbalanced T helper (Th)1 and Th17 cell response with increased production of interferon gamma (IFNγ), interleukin (IL) 17A, IL17F and IL22 [3]. Furthermore, CD has a relatively decreased regulatory T cells (Treg) response resulting in decreased production of anti-inflammatory cytokines such as transforming growth factor beta (TGF-β) and IL10 [4]. Th17 and Treg are opposing phenotypes, and their differentiations are inversely regulated by the transcription factors RORγT/FOXP3 [5]. The discussion as to whether increased IL22 levels play a beneficial role, by promoting epithelial regeneration, or contribute to the development of chronic inflammation in inflammatory bowel diseases (IBD) is ongoing [6,7].

Vitamin D is an immune modulator with proposed significant benefits in patients with autoimmunity [8]. In CD, vitamin D deficiency is associated with more frequent CD relapse rates, while vitamin D treatment increases the possibility of remission [9,10,11,12].

Vitamin D deficiency has also been associated with reduced efficacy of anti-TNFα treatment in inflammatory bowel diseases (IBD) [13,14]. It has been observed that IBD patients responding to anti-TNFα treatment have higher vitamin D levels than non-responders [15]. However, which dosage to use also considering the risk of vitamin D toxicity is still disputed [16,17].

Vitamin D may modulate the inflammatory response through the vitamin D receptor (VDR), which is present on most immune cells including T cells [18]. In vitro vitamin D reduces the Th17 cell secretion of IL-17, IL-21 and IFNγ in human T cells [19] and in mice [20].

Furthermore, vitamin D increases the plasma levels of the antimicrobial peptide cathelicidin and enhances the mucosal barrier in CD [21].

It is unknown whether high-dose vitamin D treatment can modulate the mucosal T cell immune function in active CD and to what extent combined infliximab and vitamin D treatment can increase the mucosal immune response in active CD.

We hypothesized that high-dose vitamin D treatment alone or in combination with infliximab in patients with active Crohn’s disease decreases the mucosal expression of IL22, IL17A and IL17F and increases the mucosal VDR expression. Furthermore, we hypothesized that high-dose vitamin D treatment decreases endoscopic disease activity and inflammatory markers. We conducted a randomized, double-blinded, placebo-controlled study to address these hypotheses.

## 2. Materials and Methods

### 2.1. Study Approval and Conduction

The study was a single-center study carried out at Aarhus University Hospital, Denmark from July 2014 to October 2017. The study was approved by the Danish Medicine Agency (EudraCT no. 2013-000971-34), the Central Denmark Regional Committee for Health Research Ethics (no. 1-10-72-141-13) and the Danish Data Protection Agency (no. 1-16-02-296-13). The study was conducted according to the guidelines of good clinical practice (GCP) and was monitored by the GCP unit at Aarhus University Hospital. Data were collected and documented in a patient-specific case report form (CRF). Upon completion of the study, inclusion data were transferred into Research Electronic Data Capture (REDCap) [22]. The results are posted in EudraCT.

### 2.2. Inclusion and Exclusion Criteria

Patients were provided written and oral information regarding the study. Upon receiving the patients’ signed, informed consent, they were screened according to the inclusion and exclusion criteria.

Qualifications for the enrolment of patients included the following: a diagnosis of colonic and/or ileocecal Crohn’s disease; age between 18 and 80 years; disease activity; a negative pregnancy test and written, informed consent. Disease activity was defined as a Harvey-Bradshaw Index (HBI) > 4 combined with fecal calprotectin > 100 mg/kg and/or C-reactive protein (CRP) > 8 mg/L. In addition, the endoscopic activity was examined with the Crohn’s Disease Endoscopic Index of Severity (CDEIS) score, which needed to be ≥5.

Exclusion criteria consisted of: on-going infections; tuberculosis; 25-hydroxyvitamin D2 + D3 > 100 nmol/L; treatment with biological treatment or change of azathioprine dosage within three months prior to inclusion; hypercalcaemia and/or hypercalcuria; pseudohypoparathyroidism; prior calcium-containing kidney stones; disorders of renal calcium and phosphate excretion; breastfeeding; vaccinated with a live vaccine within four weeks; untreated abscesses; oral prednisone use; or use of budesonide above 3 mg per day. Appendix A lists additional exclusion criteria regarding allergies, rare diseases and specific treatments.

### 2.3. Design and Intervention

In this double-blinded, randomized, placebo-controlled study patients were randomized to one of four groups:(1).Infliximab and vitamin D_3_ (Ifx + VitD);(2).Infliximab and placebo vitamin D_3_ (Ifx + placeboVitD);(3).Placebo infliximab and vitamin D_3_ (placeboIfx + VitD);(4).Placebo infliximab and placebo vitamin D_3_ (placeboIfx + placeboVitD).

Patients receiving vitamin D_3_ (Dekristol) received a bolus of 200,000 international units (IU) (5 mg) orally followed by an oral daily dose of 20,000 IU (0.5 mg) for the rest of the study. Patients in the placebo vitamin D_3_ group received the same number of capsules as the vitamin D_3_ treated patients. Patients randomized to infliximab received infliximab infusion with 5 mg/kg at baseline, week 2 and week 6. Patients randomized to placebo infliximab were treated with 0.9% sodium chloride infusion at the same intervals as the patients treated with infliximab. Intervention and placebo medicine were identical in appearance. Treatment was blinded to both patients and investigators.

A baseline endoscopy was performed on all patients and endoscopic disease activity was estimated by the CDEIS score. Biopsies (2 × 2 mm) from inflamed tissue were obtained for PCR analyses. HBI, calprotectin, CRP and albumin levels were measured at weeks 0, 2 and 6 (±7 days for blood samples and ±14 days for calprotectin).

Patients who developed aggravation of the disease activity (increase in HBI and/or increase in CRP or calprotectin) at week 2 underwent a rescue endoscopy at week 3 with biopsies and CDEIS score measurement, and treatment with infliximab/placebo-infliximab was unblinded. Placebo-infliximab treated patients then received rescue infliximab induction and continued infliximab treatment for the rest of the study. Patients who initially were randomized to infliximab were dose escalated to 10 mg/kg. Project treatment regarding vitamin D/placebo vitamin D in the patients with aggravated disease activity was stopped at the day of rescue endoscopy.

At week 7, patients had a final endoscopy with biopsies, and the CDEIS score was measured. Treatment response was evaluated based on changes in CDEIS, HBI, calprotectin, CRP and albumin levels. Non-responders were unblinded at week 7 regarding the infusion treatment, and 5 mg/kg of infliximab treatment was initiated if they had received placebo infliximab infusions at weeks 0, 2 and 6. Week 6 and 7 data include rescue treated patients to avoid positive selection of the less disease active patients (Figure 1).

### 2.4. Randomization and Blinding

Patients were assigned a number from 1 to 40 continuously as they were enrolled in the study. Patients were block randomized in groups of 10 by the Hospital Pharmacy at Aarhus University Hospital (AUH). The randomization was blinded for both patients and investigators.

Dekristol capsules were produced by Mibe GmbH (Arzneimittel, Brehna, Germany). Blinded vitamin D_3_ and placebo capsules were made by the hospital pharmacy at Aarhus University Hospital. The medicine was delivered in two containers. One container was for bolus treatment containing 5 capsules with a total of either 5 mg vitamin D_3_ or placebo. The other container held 49 capsules each containing either 0.5 mg vitamin D_3_ or placebo. These capsules were for daily use until the patient’s final colonoscopy. Blinding of the infusion treatment was conducted by handing a concealed envelope with the specific patient number to an unblinded nurse, who documented and prepared the infusion treatment according to the randomization.

For each randomization number, two sealed envelopes were kept at the Department of Hepatology and Gastroenterology at AUH with information regarding the study medication: one envelope for capsule treatment and one envelope for infusion treatment. In case of unexpected adverse events, both envelopes could be opened if necessary. During the study period, two capsule envelopes were opened due to side effects being experienced by two patients. One capsule envelope was opened unintentionally at rescue treatment. All project treatments were stopped at the rescue endoscopy. A total of ten infusion envelopes were opened for patients requiring rescue treatment. A total of 13 envelopes were opened within the four groups due to unsatisfying treatment response at week 7.

### 2.5. Biopsies for Quantitative Polymerase Chain Reaction (qPCR)

At endoscopy, four double biopsies (2 × 2 mm) were taken from the inflamed mucosa for qPCR analyses at week 0 and from the same area at the final endoscopy. The biopsies were directly snap frozen in liquid nitrogen and transferred to a tube and frozen at −140 °C.

### 2.6. Isolation of Ribonucleic Acid (RNA)

Sixteen hours before tissue homogenization, 500 mL of RNA*later*-ICE (Life Technologies, Carlbad, CA, USA) was added to every tube, and afterwards, the tubes were transferred to a −20 °C freezer. Biopsies were chopped twice on ice and transferred to a TissueLyser tube where 500 mL RTL buffer was added. Two 5 mm beads were added, and samples were homogenized with a TissueLyser II (Qiagen GmbH, Hilden, Germany) according to the manufacturer’s description. RNA purity was tested by measuring sample absorption at 260 and 280 nm wavelengths on a spectrophotometer and calculating the 260/280 ratio (OD), which was only accepted if above 1.8. RNA concentration and OD were measured with NanoDrop 2000 (Thermo Fisher Scientific, Wilmington, DE, USA) according to the manufacture’s description. The quality of the RNA was evaluated by performing an RNA integrity number (RIN) score using Agilent RNA 6000 Nano Kit (Agilent Technologies, Waldbronn, Germany) and a Bioanalyser. RIN score above 6.5 was accepted.

### 2.7. Quantitative PCR

RNA was transcribed to cDNA using SuperScript IV VILO (Thermo Fisher Scientific, Waltham, MA, USA); 300 ng RNA per sample was used. No template controls using H_2_O were included to indicate potential problems with non-specific amplification or sample contamination. Non-reverse transcriptase controls were included to assess potential DNA contamination. A positive control using a pool of all samples was prepared in a standard cDNA synthesis reaction. In short, 300 ng RNA was mixed with 4 μL of Superscript™ VILO™ MasterMix (Thermo Fisher Scientific, Waltham, MA, USA) diluted with diethylpyrocarbonate

(DEPC)-treated water to a total volume of 20 μL and incubated at 25 °C for 10 min, 50 °C for 10 min and 85 °C for 5 min.

Quantitative PCR was performed on a StepOnePlus™ Real-Time PCR System (Thermo Fisher Scientific, Waltham, MA, USA). Amounts of 1 µL cDNA, 10 µL TaqMan^®^ Gene Expression Master Mix 5, 8 µL UltraPure DEPC treated water (Invitrogen, Carsbad, CA, USA) and 1 µL 20x TaqMan^®^ hydrolysis probe were mixed and incubated at 50 °C for 2 min and 95 °C for 2 min followed by 40 cycles of 95 °C for 1 s and 60 °C for 20 s. Data were acquired using the StepOne Software version 2.3 (Thermo Fisher Scientific, Waltham, MA, USA). The used Taqman probes are listed in Appendix A.

All qPCR reactions were performed in duplicates. The expression value of each gene relative to the endogenous control genes (Δ*C*_q_) was calculated as 2^(^*^C^*^q^
^(target-gene) − *C*q^
^(rfg))^, where *rfg* is the mean of the reference gene RPS9. Three reference genes were tested (RPS9, B2M, HPLPO), and the variability was tested with Normfinder, showing that RSP9 alone had the lowest variability on 0.043.

### 2.8. Vitamin D Safety Markers

Serum 25-OH vitamin D_2_ + D_3_ was analyzed using high-performance liquid chromatography-tandem mass spectrometry on mass spectrometers API3000™ or API5500™ (Applied Biosystems, Lincoln, OR, USA). An amount of 1,25(OH)_2_ vitamin D_3_ was measured using liquid chromatography–mass spectrometry and detected by Electrospray Ionization Tandem Mass Spectrometry. The 5500 Triple Quadrupole or 6500 QTRAP (AB Sciex, Framingham, MA, USA) were used for these analyses. Serum ionized calcium was measured by a potentiometric method on a Nova CRT 8 electrolyte analyzer (Diamond Diagnostics Inc., Holliston, MA, USA). Plasma PTH was measured by an immunometric assay and plasma phosphate by absorption photometry, both on a Cobas^®^ 6000 (Roche Diagnostics, Indianapolis, IN, USA).

### 2.9. Statistical Analyses

The repeated measurement data were analyzed using a mixed model. Patients were included as a random effect. An unstructured error variance-covariance matrix was chosen to allow for possible differences in correlations and standard deviations between measurements corresponding to different visits. After inspection of plots of standardized residuals versus fitted values and QQ-plots of the standardized residuals, analysis was performed on all measurements using a logarithmic scale. Results are given as estimated medians (back-transformed means on the logarithmic scale) with 95%-confidence intervals. Groups were compared using ratios of estimated medians (back-transformed mean differences) with 95% confidence intervals. Patient characteristic data were analyzed with Prism 6 (GraphPad Software, Inc., La Jolla, CA, USA). The repeated measurements data were analyzed using Stata version 15.1.

Sample size: Exact sample size calculation in this study was not possible as no reference interval regarding mucosal cytokine expressions, during vitamin D treatment to patients with active CD was reported.

## 3. Results

### 3.1. Patient Characteristics and Study Follow-Up

Fifty-nine patients with active CD were informed about the study (Figure 1). Nine patients declined to participate. Ten patients provided written, informed consent but were excluded according to the inclusion and exclusion criteria. We randomized forty patients to trial treatment. One patient suffered a severe infusion reaction to infliximab during the second infusion and was excluded from the study prior to having the final colonoscopy. Thirty-nine patients were examined with the final colonoscopy. Ten out of 40 patients underwent rescue treatment at week 3.

Table 1 shows the baseline characteristics of the four groups. Despite randomization of the participants, the Ifx + placeboVitD group included more azathioprine users compared to the other three groups (Table 1). The following four severe adverse events were documented during the study: (1) one patient experienced a severe infusion reaction to infliximab (Ifx + VitD group); (2) two patients were hospitalized due to IBD-related surgery (Ifx + placeboVitD and placeboIfx + placeboVitD group); and (3) one patient was kept for observation for a possible allergic reaction to trial treatment (Ifx + VitD group). A full list of documented adverse events (AE) during the intervention appears in Appendix A of the Appendix A.

### 3.2. High-Dose Vitamin D and Infliximab Decrease Mucosal Th17-Related Cytokine Expression

Seven weeks of high-dose vitamin D as monotherapy decreased the mucosal IL17A expression by 55% (median ratio 0.45 (95%CI: 0.22–0.95)) (*p* = 0.04), with no significantly changes in the placebo group (median ratio 0.61 (95%CI: 0.19–1.94)) (Table 2). Ifx + VitD also decreased IL17A expression with 81% (median ratio 0.19 (95%CI: 0.07–0.55)) (*p* = 0.002), while monotherapy with infliximab tended to reduce the IL17A expression (median ratio 0.39 (95%CI: 0.14–1.07)) (*p* = 0.07). However, the changes in IL17A mRNA expression did not reach significance between the groups.

IL-22 and IL-17F expression were affected by infliximab and not significantly by high-dose vitamin D. The Ifx + VitD group decreased the IL22 expression with 97%(median ratio 0.03 (95%CI: 0.005–0.12)) (*p* = 0.0001) and the IL17F expression with 74% (median ratio 0.25 (95%CI: 0.11–0.6)) (*p* = 0.002), while the Ifx + placeboVitD group reduced the IL22 expression with 89% (median ratio 0.12 (95%CI: 0.03–0.46)) (*p* = 0.002) and the IL17F expression with 64% (median ratio 0.36 (95%CI: 0.16–0.81)) (*p* = 0.013) (see Table 2). The decreases in IL22 and IL17F expressions were not significantly different between the two groups. Monotherapy with high-dose vitamin D or placebo-placebo did not significantly decrease the IL22 (placeboIfx + VitD: median ratio 0.47 (95%CI: 0.17–1.26, *p* = 0.13), placeboIfx + placeboVitD: median ratio 0.57 (95%CI: 0.12–2.62, *p* = 0.47)) or IL17F expressions (placeboIfx + VitD: median ratio 0.92 (95%CI: 0.5–1.69, *p* = 0.78), placeboIfx + placeboVitD: median ratio 0.97 (95%CI: 0.39–2.44, *p* = 0.95)). VDR expression was not significantly affected by treatment (see Table 2).

### 3.3. High-Dose Vitamin D Treatment Decreases IL10 and INFγ Expression

To further describe the vitamin D-related modulation of the mucosal immune response, the mucosal mRNA expression of IL-10, IFNγ, TNFα and TGFβ was examined together with the vitamin D-related cathelicidin (CAMP) and 1α-hydroxylase (CYP27B1). Vitamin D treatment alone and in combination with infliximab reduced the expression of IL10 with 37% (median ratio 0.63 (95%CI: 0.43–0.93)) (*p* = 0.02) and 50% (median ratio 0.49 (95%CI: 0.27–0.88)) (*p* = 0.02), respectively. There were no significant changes in the IL-10 expression in the Ifx + placeboVitD group or in the placeboIfx + placeboVitD group. However, changes between groups did not reach significance (Table 3). IFNγ expression was decreased significantly both by vitamin D and infliximab treatment but not by placebo-placebo. Seven weeks with vitamin D monotherapy reduced the IFNγ expression with 53% (median ratio 0.47 (95%CI: 0.23–0.95)) (*p* = 0.03). In the Ifx + VitD group IFNγ, expression was reduced by 79% (median ratio 0.21 (95%CI: 0.07–0.59)) (*p* = 0.003) and in the Ifx + placeboVitD group by 63% (median ratio 0.37 (95%CI: 0.14–1.0)) (*p* = 0.05). TNFα expression was reduced in the Ifx + placeboVitD group with 55% (median ratio 0.47 (95%CI: 0.25–0.85)) (*p* = 0.01) and vitamin D monotherapy tended to reduce TNFα with 32% (median ratio 0.68 (95%CI: 0.45–1.03)) (*p* = 0.07). Vitamin D did not significantly change the expression of CAMP or CYP27B1, but the expression of CAMP was reduced significantly by infliximab in the Ifx + VitD group (median ratio: 0.39 (95%CI: 0.16–0.93) (*p* = 0.03)), and CYP27B1 was reduced in both infliximab groups (Ifx + VitD median ratio: 0.03 (95%CI: 0.005–0.14), Ifx + placeboVitD median ratio: 0.05 (95%CI: 0.01–0.2)) (*p* < 0.0001).

### 3.4. High-Dose Vitamin D Treatment Did Not Decrease Endoscopic Disease Activity

Vitamin D treatment did not result in significant differences in CDEIS score changes between the placeboIfx + VitD and placeboIfx + placeboVitD groups (median ratio: 0.9 (95% CI: 0.5–1.9)) (*p* = 0.8) or the Ifx + VitD versus Ifx + placeboVitD group (median ratio: 1.7 (95% CI: 0.7–4.1)) (*p* = 0.21) (Figure 2A).

Overall changes in CDEIS scores between the four groups were dependent on the given project treatment (mixed model test for parallel curves, *p* < 0.0001). The CDEIS scores of the two infliximab groups declined more than in the placeboIfx + VitD and placeboIfx + placeboVitD groups (*p* < 0.02).

Vitamin D treatment did not significantly affect changes in HBI score in the placeboIfx + VitD versus placeboIfx + placeboVitD groups (mean difference: 0 (95%CI: −2.5–2.5)) (*p* = 1.0) or in the Ifx + VitD versus Ifx + placeboVitD groups (mean difference: −0.8 (95%CI: −3.8–2.2)) (*p* = 0.61). Similar to the CDEIS scores, the HBI scores of the two infliximab groups declined more compared to the placeboIfx + VitD (*p* < 0.03) and placeboIfx + placeboVitD groups (*p* = 0.05) (Figure 2B).

### 3.5. No Significant Effect of Vitamin D on Calprotectin and CRP

Changes in calprotectin and CRP levels were not affected significantly by vitamin D treatment. The calprotectin levels of the placeboIfx + VitD group did not decline significantly compared to the placeboIfx + placeboVitD group (median ratio: 0.71 (95% CI: 0.28–1.84)) (*p* = 0.48). Similarly, the Ifx + VitD group did not decline significantly more than the Ifx + placeboVitD group (median ratio: 0.45 (95% CI: 0.14–1.39)) (*p* = 0.17) (Figure 3A). Calprotectin levels declined more in the two infliximab groups compared to the placeboIfx + VitD and placeboIfx + placeboVitD groups (*p* < 0.03). This decline resulted in differences between overall calprotectin levels between the four groups (mixed model for parallel curves, *p* = 0.01).

CRP levels in the placeboIfx + VitD group were comparable to the placeboIfx + placeboVitD group (median ratio: 1.17 (95%CI: 0.59–2.30)) (*p* = 0.66) (Figure 3B). Likewise, the Ifx + VitD group did not decline significantly more than the Ifx + placeboVitD group (median ratio: 1.15 (95%CI: 0.52–2.54)) (*p* = 0.73). The differing changes in CRP levels between the four groups (mixed model test for parallel curves, *p* = 0.01) were driven by a greater CRP decline in the two infliximab groups compared to the placeboIfx + VitD and placeboIfx + placeboVitD groups (*p* < 0.03).

Albumin concentrations did not differ significantly between the groups over time (data not shown).

### 3.6. High-Dose Vitamin D Treatment Is Safe

Patients randomized to vitamin D treatment received a maximum of 29.5 mg vitamin D during 7 weeks, and as expected the 25-vitD levels were dependent on actual vitamin D treatment (*p* < 0.0001, test for parallel curves) (Figure 4). During active vitamin D treatment, the levels of 25-vitD increased in the Ifx + VitD and placeboIfx + VitD groups from week 0 to 6 with a 4.6 factor (95% CI: 3.4–6.2) (median 192 nmol/L, 95%CI: 152–233) and 3.4 factor (95% CI: 2.7–4.2) (median 218.7 nmol/L, 95%CI: 187–251), respectively. Similarly, the 1.25-vitD levels were dependent on project treatment (*p* < 0.0001, test for parallel curves) (Figure 4B). Although vitamin D treatment significantly increased the calcium-ion levels (*p* = 0.01, mixed model test for parallel curves), the induction was within the reference range (Figure 4C). Parathyroid hormone levels were affected by vitamin D treatment (test for parallel curves, *p* = 0.018) with reduced levels within the reference range in the vitamin D-treated groups (Figure 4D). None of the patients experienced hyperphosphatemia (data not shown).

## 4. Discussion

In the present study, we investigated if high-dose vitamin D treatment alone or combined with infliximab changed the mucosal cytokine mRNA expression compared to placebo and whether vitamin D treatment reduced the disease activity in CD. High-dose vitamin D and infliximab treatment decreased the mucosal expression of IL17A. Furthermore, IL-10 and IFNγ expression was decreased by vitamin D treatment. However, the vitamin D-induced reduction in cytokine expression was not reflected in reduced disease activity scores or endoscopic improvements, neither decreased serological nor fecal inflammatory markers.

Seven weeks of high-dose vitamin D treatment alone or combined with infliximab decreased the mucosal mRNA IL17A expression within groups. This indicates a beneficial role of vitamin D in CD as IL17 is considered to be a pivotal proinflammatory cytokine in CD inflammation [23,24]. Our finding is supported by in vitro studies showing that stimulation of CD4 T cells with 1.25-vitD decreases the expression of IL17A, IL17F and IL22 [25,26] perhaps by specific VDR binding to the IL17A promotor, resulting in reduced IL17A production [26]. However, IL17A may also have beneficial roles in CD since treatment with anti-IL-17A increased the disease activity in a subgroup of patients with elevated CRP compared to placebo [27].

The observed vitamin D-related decrease in IL-10 expression, an anti-inflammatory cytokine, is somewhat unexpected. One would anticipate that the anti-inflammatory vitamin D effect should result in increased expression of the anti-inflammatory cytokine IL-10. Cross sectional studies have indicated that high 25-vitD levels were associated with increased circulating and mucosal expression of IL-10 in IBD patients [28,29]. However, our study is the first to investigate the individual changes in mucosal IL10 expression from high-dose vitamin D, and our result does not support that vitamin D treatment is mediated through IL-10. The reduced IL-10 levels could be interpreted as an indirect consequence of reduced levels of proinflammatory IL-17A, IFNγ and TNF-α requiring reduced production of anti-inflammatory IL-10, since the order of the regulatory steps during the treatment are unknown.

Although monotherapy with high-dose vitamin D did affect several mucosal cytokine expressions, we did not observe significant reductions in endoscopic disease activity, HBI score or biochemical inflammatory markers compared to placebo. In addition, Ifx + VitD treatment was not significantly superior to infliximab as monotherapy. In this perspective, our study seems not to support that high-dose vitamin D treatment alone has a clinically significant effect on active CD. However, it should be emphasized that this study only had a small number of patients randomized to four groups, as it was designed to investigate the possible mucosal effects of vitamin D treatment in CD, and was underpowered to investigate the possible clinical effects.

It has previously been suggested that vitamin D improves the treatment response of infliximab in CD patients [13,14]. However, a study of IBD patients using low doses of vitamin D supplemented with 150,000 IU vitamin D every 3 months did not influence disease activity through a 12-month follow-up [30].

In the present study, we chose a high vitamin D dosing of 29.5 mg for 7 weeks to ensure that the result was not biased by a dose–response phenomenon. This dosage was well-tolerated in CD patients with disease activity. We have previously demonstrated that healthy subjects also tolerate high-dose vitamin D treatment without developing hypercalcemia [31]. In the present study, high-dose vitamin D treatment alone or combined with infliximab did not result in a greater number of adverse events than treatment with placebo or infliximab alone.

Our study does have limitations. The sample size was relatively small. This resulted in relatively wide confidence intervals for cytokine expression in the groups, especially those with only eight patients. Consequently, the study is probably underpowered to demonstrate minor differences in cytokine expression changes between the different treatment groups. This could at least partly explain why we did not observe significant differences in cytokine expression changes between the vitamin D-treated groups and the placebo-vitamin-treated groups. Although forty CD patients were randomized into four groups, the infliximab monotherapy group inadvertently had more azathioprine users than the other three groups. This could result in higher anti-inflammatory therapy load in this group. However, the monoinfliximab group did not demonstrate superior treatment effects compared to the combined infliximab and vitamin D group. Furthermore, vitamin D levels are not influenced by the use of azathioprine [32,33]. To avoid selection bias, we combined data from rescue-treated patients (week 3) with week 7 data from patients who followed the full protocol. The patients who underwent rescue treatment only received two infliximab infusions and 3 weeks of capsule treatment, which might influence the results.

## 5. Conclusions

In this double-blinded, randomized, controlled study, high-dose vitamin D alone and combined with infliximab reduced the mucosal expression of IL17A, IFNγ and IL10 within groups. However, high-dose vitamin D treatment did not significantly decrease the observed clinical disease activity during the 7 weeks of intervention.

## Figures and Tables

**Figure 1 nutrients-12-03699-f001:**
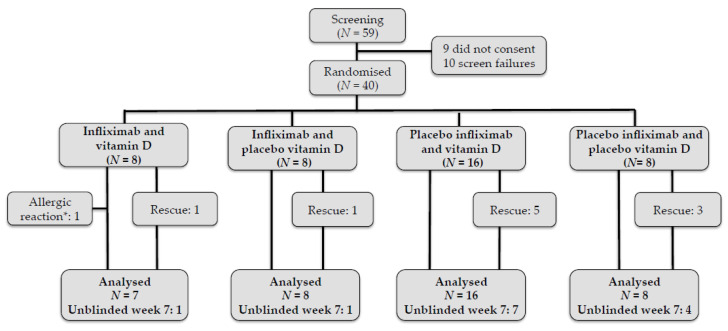
Patient flow during the 7-week study period. Patients who underwent rescue treatment were included in the final analyses. * One patient had an allergic reaction to infliximab at the second infusion and was excluded. Infusion treatment in non-responding patients with on-going endoscopic inflammation was unblinded at week 7, and infliximab treatment was initiated.

**Figure 2 nutrients-12-03699-f002:**
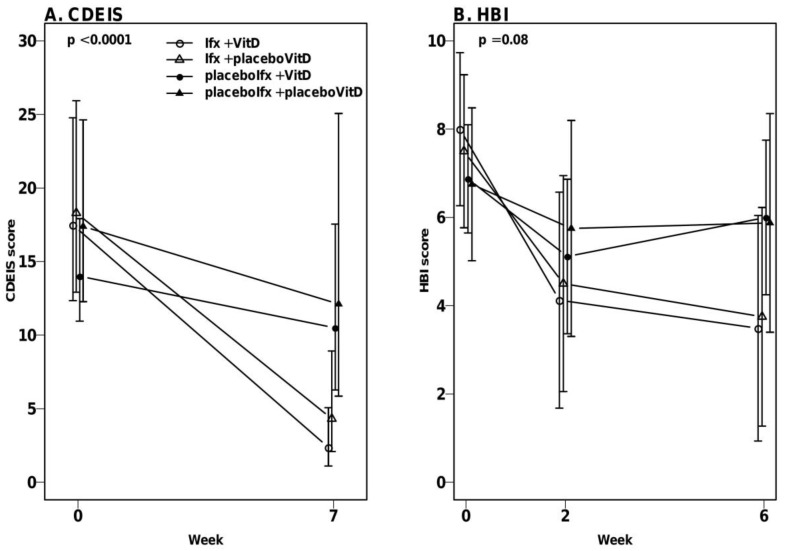
Crohn’s Disease Endoscopic Index of Severity (CDEIS) and Harvey-Bradshaw Index (HBI) scores during the active vitamin D treatment. Dots represent medians with 95% CI. (**A**). CDEIS scores at weeks 0 and 7 (“week 7” includes week 3 rescue colonoscopy) within the four groups. Vitamin D treatment did not affect changes in CDEIS score in neither the Ifx + VitD versus Ifx + placeboVitD groups nor in the placeboIfx + VitD versus placeboIfx + placeboVitD groups. The CDEIS score of both infliximab groups declined significantly compared to the placeboIfx + VitD and placeboIfx + placeboVitD group (*p* < 0.02). (**B**). HBI scores at weeks 0, 2 and 6 (week 6 includes rescue week 3 HBI scores). HBI changes were not influenced by vitamin D treatment in either the Ifx + VitD versus Ifx + placeboVitD groups or in the placeboIfx + VitD versus placeboIfx + placeboVitD groups. Similar to the CDEIS scores, the HBI scores of the two infliximab groups declined in comparison to the placeboIfx + VitD (*p* < 0.03) and placeboIfx + placeboVitD groups (*p* = 0.05). VitD, vitamin D; Ifx, Infliximab.

**Figure 3 nutrients-12-03699-f003:**
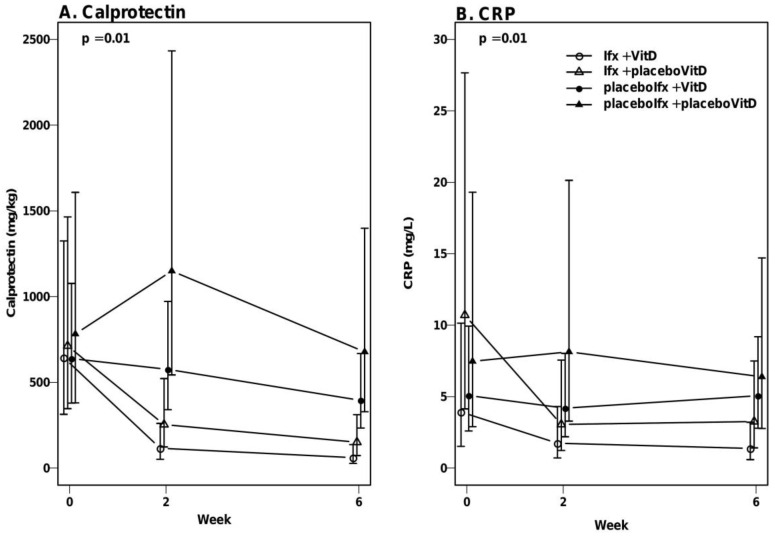
Calprotectin and CRP levels during active vitamin D treatment. Week 6 includes data from both patients who underwent rescue treatment (week 3) as well as patients who followed the protocol until week 6. The dots represent median with 95% CI (bars). (**A**). Changes in calprotectin levels over time are significantly different between the groups. These changes are a direct result of infliximab treatment (*p* = 0.01). No significant differences were observed between Ifx + VitD and Ifx + placeboVitD (*p* = 0.17) or between placeboIfx + VitD and placeboIfx + placeboVitD (*p* = 0.48). (**B**). Changes in CRP levels over time are significantly different between the groups as a result of the infliximab and not the vitamin D treatment (*p* = 0.01). Only the Ifx + VitD group (not the Ifx + placeboVitD group) demonstrated a more significant decline in CRP levels compared to the placeboIfx + VitD and placeboIfx + placeboVitD groups. VitD, vitamin D; Ifx, Infliximab.

**Figure 4 nutrients-12-03699-f004:**
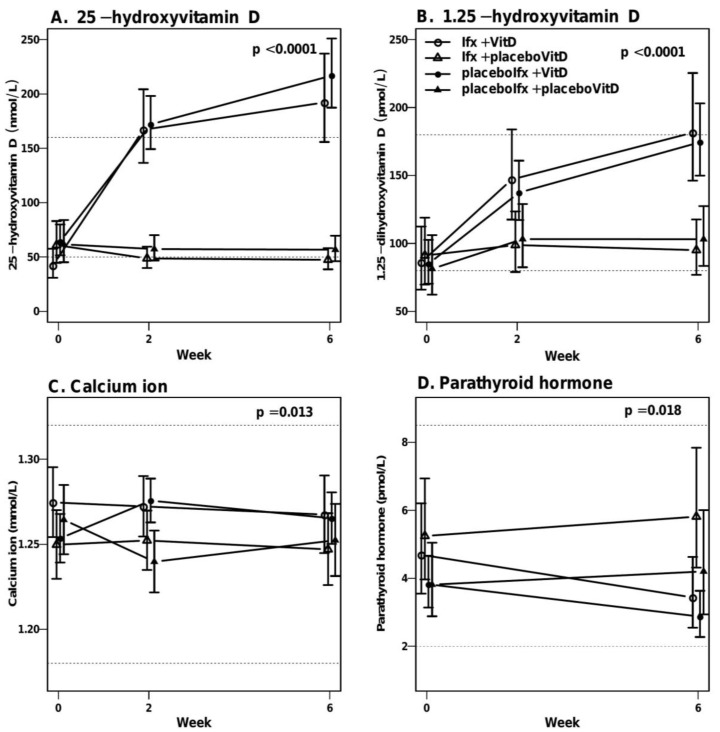
Vitamin D, calcium and parathyroid levels from week 0 to 6. “Week 6” also comprises rescue treated patients (week 3). A mixed model was used to test for parallel curves. Data are presented as medians with 95% CI. Dotted lines represent the reference range. (**A**). Changes in 25-hydroxyvitamin D_2_ + D_3_ (25-vitD) over time were significantly different in the four groups (*p* < 0.0001) as expected. (**B**). Changes in active 1.25-dihydroxyvitamin D_3_ (1.25-vitD) were also dependent on treatment (*p* < 0.0001) with higher 1.25-vitD levels in the vitamin D treated groups. (**C**). Changes in calcium-ion levels over time were dependent on treatment (*p* = 0.01), again with the highest levels in the vitamin D treated groups. However, all patient levels were within the reference range. (**D**). Changes in parathyroid hormone levels over time were dependent on treatment (*p* = 0.018), though all patient levels were within the reference range. VitD, vitamin D; Ifx, Infliximab.

**Table 1 nutrients-12-03699-t001:** Baseline patient characteristics.

	Ifx + VitD	Ifx + PlaceboVitD	Placeboifx + VitD	PlaceboIfx + placebovitd
*n*	8	8	16	8
BMI	25.0 (21.5–29.9)	22.8 (20.5–44.3)	24.8 (18.7–32.9)	28.8 (23.1–34.2)
Years since diagnosis	2.3	3.3	1.2	1.9
Smoking	3 (37.5%)	1 (12.5%)	1 (6.3%)	3 (37.5%)
Family history of CD or UC	1	2	3	1
Former or present stenosis	3 (37.5%)	3 (37.5%)	1 (6.3%)	3 (37.5%)
Former abdominal surgery	1 (12.5%)	0	3 (18.8%)	1 (12.5%)
Former or present fistula	0	1 (12.5%)	1 (6.3%)	0
Former or present abscess	0	2 (25%)	2 (12.5%)	1 (12.5%)
Former or present fissure	0	0	1 (6.3%)	0
Other autoimmune diseases	0	2 (25%)	2 (12.5%)	1 (12.5%)
Extra intestinal manifestations	2 (25%)	2 (25%)	5 (31.3%)	0
Fatigue	4 (50%)	5 (62.5%)	13 (81.25%)	4 (50%)
Former treatment with infliximab	2 (25%)	3 (37.5%)	4 (25%)	1 (12.5%)
Former treatment with Adalimumab	0	0	2 (12.5%)	1 (12.5%)
Former treatment with other biologicals	0	0	1 (6.3%)	0
Budesonide users (3 mg/day)	0	1 (12.5%)	0	0
Azathioprine users	3 (37.5%)	6 (75%)	3 (18.8%)	1 (12.5%)
HBI	7 (5–14)	6.5 (5–16)	7 (5–11)	5 (5–10)
calprotectin, mg/kg	884 (114–2174)	718 (163–3366)	895 (113–2094)	714 (256–6000)
25-hydroxyvitamin D, nmol/L	45 (11–83)	73 (33–88)	66.5 (32–94)	72.5 (21–90)
CRP, mg/L	5.3 (0.6–25.6)	16.5 (0.6–38.5)	8.8 (0.3–25)	6.3 (0.8–45.9)
Leucocytes, 10^9^/L	7.67 (5–11.6)	7.1 (5.6–9.4)	7.6 (5.0–11.5)	8.2 (3.7–15.8)
Haemoglobin, mmol/L	8.7 (7.5–10.2)	8.6 (6.4–9.5)	8.5 (6.9–9.7)	8.5 (6.6–10)
SH score	21 (11–38)	20 (3–33)	14.5 (4–36)	21 (5–33)
Endoscopic CDEIS score	13 (11–32)	18 (7–33)	14 (6–29)	19 (6–49)

Patient characteristic. Data are given in median with range or a total number with percentage. CD—Crohn’s disease; UC—ulcerative colitis; BMI—body mass index; HBI—Harvey-Bradshaw Index; CRP—C-reactive protein; SH score—short health score; CDEIS score—Crohn’s disease index of severity score.

**Table 2 nutrients-12-03699-t002:** Changes in mucosal mRNA expression of IL22, vitamin D receptor (VDR), IL17F and IL17A.

	1. Ifx + VitD		2. Ifx + PlaceboVitD		Group 1 vs. 2	3. PlaceboIfx + VitD		4. PlaceboIfx + PlaceboVitD		Group 3 vs. 4
	Week 0: *N* = 7 Week 7: *N* = 7		Week 0: *N* = 7 Week 7: *N* = 8			Week 0: *N* = 14 Week 7: *N* = 16		Week 0: *N* = 8 Week 7: *N* = 5		
Gene	Median ratio (95%CI)	*p*	Median ratio (95%CI)	*p*	*p*	Median ratio (95%CI)	*p*	Median ratio (95%CI)	*p*	*p*
IL22	0.03 (0.005–0.12)	0.00001	0.12 (0.03–0.46)	0.002	0.15	0.47 (0.17–1.26)	0.13	0.57 (0.12–2.62)	0.47	0.83
VDR	0.7 (0.34–1.45)	0.34	0.83 (0.42–1.67)	0.61	0.34	1.37 (0.84–2.22)	0.21	0.72 (0.32–1.62)	0.42	0.18
IL17F	0.25 (0.11–0.6)	0.002	0.36 (0.16–0.81)	0.013	0.55	0.92 (0.5–1.69)	0.78	0.97 (0.39–2.44)	0.95	0.92
IL17A	0.19 (0.07–0.55)	0.002	0.39 (0.14–1.07)	0.07	0.34	0.45 (0.22–0.95)	0.04	0.61 (0.19–1.94)	0.40	0.67

Changes in mucosal mRNA expression from week 0 to 7. Results are given as median ratio (week 7/week 0) with 95% CI. Median ratios are compared for group 1 versus 2 and for group 3 versus 4 respectively. *N* was reduced in some of the groups due to low RNA quality or no detectable cDNA expression. All analyses were conducted using a mixed model, which can take missing values into account. IL4 expression was also tested but was not detectable. *N*, number; Ifx, infliximab; VitD, vitamin D; VDR, vitamin D receptor; IL, interleukin.

**Table 3 nutrients-12-03699-t003:** Changes in mucosal mRNA expression of from week 0 to 7.

	1. Iflx + VitD		2. Iflx + placeboVitD		Group 1 vs. 2	3. PlaceboIfx + VitD		4. PlaceboIfx + PlaceboVitD		Group 3 vs. 4
	Week 0: *N* = 7 Week 7: *N* = 7		Week 0: *N* = 7 Week 7: *N* = 8			Week 0: *N* = 14 Week 7: *N* = 16		Week 0: *N* = 8 Week 7: *N* = 5		
Gene	Median ratio (95%CI)	*p*	Median ratio (95%CI)	*p*	*p*	Median ratio (95%CI)	*p*	Median ratio (95%CI)	*p*	*p*
TGF-β	0.82 (0.45–1.52)	0.53	0.73 (0.41–1.29)	0.28	0.77	0.70 (0.47–1.05)	0.09	1.17 (0.59–2.31)	0.65	0.21
IL10	0.49 (0.27–0.88)	0.02	0.75 (0.43–1.3)	0.31	0.3	0.63 (0.43–0.93)	0.02	0.92 (0.48–1.75)	0.80	0.33
IFNγ	0.21 (0.07–0.59)	0.003	0.37 (0.14–1.00)	0.05	0.43	0.47 (0.23–0.95)	0.03	0.96 (0.30–3.02)	0.94	0.3
TNF-α	0.61 (0.32–1.13)	0.12	0.47 (0.25–0.85)	0.01	0.55	0.68 (0.45–1.03)	0.07	1.20 (0.59–2.45)	0.62	0.18
CAMP	0.39 (0.16–0.93)	0.03	0.49 (0.21–1.12)	0.09	0.71	1.05 (0.59–1.86)	0.88	1.26 (0.47–3.37)	0.64	0.75
CYP27B1 *	0.03 (0.005–0.14)	0.0001	0.05 (0.01–0.20)	0.0001	0.57	0.52 (0.20–1.39)	0.2	1.16 (0.25–5.33)	0.84	0.39

Changes in mucosal mRNA expression from week 0 to 7. Results are given as median ratio (week 7/week 0) with 95% CI. Median ratios are compared for group 1 versus 2 and for group 3 versus 4, respectively. *N* was reduced in some of the groups due to low RNA quality or no detectable cDNA expression. All analyses were conducted using a mixed model, which can take missing values into account. Ifx, infliximab; vitD, vitamin D; TGF-β, transforming growth factor beta; IL, interleukin; TNF-α, tumor necrosis factor alpha; IFNγ, interferon gamma; CAMP, cathelicidin antimicrobial peptides; CYP27B1, cytochrome p450 family 27 subfamily B member 1. * Reduced numbers during low expression: group 1 week 7 *N* = 4, group 2 week 7 *N* = 6, group 3 week 7 *N* = 15.

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
