# Peer review of "Decrease in Mucosal IL17A, IFNγ and IL10 Expressions in Active Crohn’s Disease Patients Treated with High-Dose Vitamin D Alone or Combined with Infliximab"

_nutrients, 2020, doi:10.3390/nu12123699_

Round 1

Reviewer 1 Report

I had the pleasure of reviewing your article "High dose Vitamin D alone or comined with infliximab treatment decreases mucosal IL17A, INF-γ and IL-10 expressions in active Crohn's disease patients - a randomized double-blinded clinical trial". I have read the manuscript with interest and it is a well designed randomized placebo clinical trial that aims to investigate whether high doses of vitamin D with or without IFX modulate the expression of cytokines in the intestinal mucosa and if this modulation has a significant clinical relevance. Congratulations, it is a very well designed study and it shows very
interesting data.

However, I am going to make a few minor suggestions:

- Notice that you write interferon gamma sometimes as IFN and sometimes
as INF. It should be unified.
- I suggest stating in the exclusion criteria or in Table S1 that the
existence of tuberculosis has been ruled out in participating patients
(regardless of whether the "on going infections" criterion may include
it).

Author Response

Reply for Reviewer 1. 

Dear reviewer 1. Thank your for the positive comments and your suggestions. We have revised the paper according to your suggestions:  

1) Interferon gamma abbreviations are unified to IFNγ. 

2) all patients were tested for tuberculosis before inclusion. at page 2 line 85 we have included "tuberculosis" under exclusion criteria. 

All changes are coloured with red.

Reviewer 2 Report

The biological treatment is considered nowadays as the best one in dealing with IBD, both in children and adults. But still it is not enough to cure the disease. That is why researchers are trying to improve the treatment outcome by adding additional medicines.

To my understanding, that is why the authors of the paper entitle: High dose vitamin D alone or combined with 2infliximab treatment decreases mucosal IL17A, INF-g3and IL10 expressions in active Crohn’s disease 4patients -a randomized double-blinded clinical trial, having previous experience with wit. D, have investigated the concept of high dose of Vit D, as beneficial one. In my opinion this is interesting approach, since we know that Vit. D, which actually is a steroid hormone, is able to modulate the inflammatory response. The authors also are aware of the possible toxicity of Vitamin D.

The study was planned correctly also including the possible toxic effect of Vit. D by measuring concentration of 25-OH D and calcium status.

The biopsies taken from the inflamed mucosa, as a material for investigation proinflammatory interleukins activity also was proper choice and could be consider as a modern approach for the assessing inflammatory processes in the gut.

The topic of the article in my opinion is very important. The text is written clearly. The conclusions are consistent with evidence and arguments presented in the paper.

Minor revision is addressed only to 31 position of the references, which needs to add some details of cited paper( volume, pages)

Author Response

Reply for Reviewer 2:

Dear reviewer 2

Thank you for your comments, we are pleased that you found the paper interesting. we have revised the manuscript according to your suggestion: 

page 15 line 518: In reference 31 volume and pagenumber are added and the changes are highlighted in red

Reviewer 3 Report

The authors evaluated the effect of Vitamin D treatment on Crohn’s disease (CD) activity. The authors provided sufficient background and a relevant question. The study design was appropriate and methods described adequately, although the sample size seems to be relatively small. However, the results failed to reach statistical significance.

Major comment:
1. High-dose vitamin D treatment induced the reduction of IL17A, INF-γ, and IL10 expression in the endoscopic biopsy samples at week 7 compared with those at week 0. However, the reduction of the expression level in the high-dose vitamin D treatment was not statistically significant compared with those in the placebo group. Thus, Title and conclusion should be changed, because the result failed to confirm the effect of high-dose vitamin D treatment on CD mucosal immunity.

2. qPCR for selected cytokine was insufficient to evaluate the cytokine network and signaling pathway. I recommended performing the genome-wide expression pattern using RNA-seq or single-cell RNA-seq.
